# Mpp23Aa/Xpp37Aa Insecticidal Proteins from *Bacillus thuringiensis* (Bacillales: Bacillaceae) Are Highly Toxic to *Anthonomus grandis* (Coleoptera: Curculionidae) Larvae

**DOI:** 10.3390/toxins15010055

**Published:** 2023-01-08

**Authors:** Jéssica A. de Oliveira, Bárbara F. Negri, Patricia Hernández-Martínez, Marcos F. Basso, Baltasar Escriche

**Affiliations:** 1Laboratório de Prospecção de Cepas e Genes, Instituto Mato-Grossense do Algodão (IMAmt), Rondonópolis 78740-970, Mato Grosso, Brazil; 2Laboratório de Biologia Molecular e Transformação de Plantas, Instituto Mato-Grossense do Algodão (IMAmt), Rondonópolis 78740-970, Mato Grosso, Brazil; 3Departamento de Genética, Instituto de Biotecnología y Biomedicina (BIOTECMED), Universitat de València, 46100 Burjassot, Valencia, Spain; 4Dipartimento di Biologia e Incubatore Universitario Fiorentino, Dipartimento di Biologia, Università degli Studi di Firenze, 50019 Sesto Fiorentino, Firenze, Italy

**Keywords:** cotton boll weevil, Cry proteins, Mpp/Xpp proteins, Cry23Aa, Cry37Aa, Coleoptera, Curculionidae

## Abstract

The beetle *Anthonomus grandis* Boheman, 1843, is the main cotton pest, causing enormous losses in cotton. The breeding of genetically modified plants with *A. grandis* resistance is seen as an important control strategy. However, the identification of molecules with high toxicity to this insect remains a challenge. The susceptibility of *A. grandis* larvae to proteins (Cry1Ba, Cry7Ab, and Mpp23Aa/Xpp37Aa) from *Bacillus thuringiensis* Berliner, 1915, with toxicity reported against Coleopteran, has been evaluated. The ingestion of different protein concentrations (which were incorporated into an artificial diet) by the larvae was tested in the laboratory, and mortality was evaluated after one week. All Cry proteins tested exhibited higher toxicity than that the untreated artificial diet. These Cry proteins showed similar results to the control Cry1Ac, with low toxicity to *A. grandis,* since it killed less than 50% of larvae, even at the highest concentration applied (100 μg·g^−1^). Mpp/Xpp proteins provided the highest toxicity with a 0.18 μg·g^−1^ value for the 50% lethal concentration. Importantly, this parameter is the lowest ever reported for this insect species tested with *B. thuringiensis* proteins. This result highlights the potential of Mpp23Aa/Xpp37Aa for the development of a biotechnological tool aiming at the field control of *A. grandis*.

## 1. Introduction

Cotton (*Gossypium hirsutum* L.) is one of the major crops of the global agricultural economy. During the 2020/2021 growing season, more than 24 million tons of cotton fiber were produced worldwide, with an output of more than 25 million tons in the 2021/2022 growing season. Brazil is the world’s fourth-largest cotton producer after China, India, and the USA, with an expected increase of around 19% for the 2021/2022 growing season [1,2]. One of the greatest challenges for cotton cultivation in Brazil is the high demand for agricultural inputs, particularly insecticides. Data indicate that the investment in insecticides, mainly to control the *Anthonomus grandis* Boheman, 1843 (cotton boll weevil, CBW) accounts for approximately 21% of the costs of cotton farming in Mato Grosso, Brazil [3].

The cotton boll weevil is a Coleopteran insect, considered the most harmful cotton pest, and its stereotypic behavior makes control difficult [4]. In addition to feeding on the flower buds, after mating, females lay eggs inside the fruiting structures of the cotton plant, where the development of all immature stages of the insect occurs, causing abscission or reduction in fiber quality [5]. In addition, newly emerged or old adults are commonly protected by the flower bud bracts or refugees in plants outside of the cotton crops, decreasing exposure to mortality factors, i.e., insecticides applied by spraying [6].

Once the insect is detected in pheromone traps, insecticide applications are performed, using mainly malathion. These applications can be performed weekly until the insect is no longer detected. Consequently, the field may receive as many as 25 applications in a single growing season, boosting expenses regarding agricultural inputs, and negatively affecting the environment and entomofauna [7,8,9]. However, the low effectiveness of the available synthetic agents has contributed to inefficient control measures, an increase in the insect population, and its genetic diversity. In addition, the non-existence of conventional or transgenic commercial cultivars with some resistance to CBW has stimulated the search for new biotechnological tools for the effective control of this insect pest.

The development of genetically modified crops expressing Cry and Vip toxins from *Bacillus thuringiensis* (Bt) has been widely studied due to the toxic effect of these proteins, mainly against Lepidoptera [10]. Interestingly, transgenic plants expressing insecticidal proteins from Bt are effective in controlling stem borers, ear feeders, and rootworms [10]. Therefore, cotton plants expressing Bt proteins (toxic to CBW) may also control this devastating insect pest. Currently, transgenic cotton plants expressing *cry* and *vip* genes to control Lepidoptera are already being commercially launched in Brazil, but to date, no variety or cultivar capable of controlling the beetle *A. grandis* has been released [11,12,13].

The insect order Coleoptera comprises numerous species considered pests for the world’s major crops, few of which have proved susceptible to Bt toxins [14]. Despite causing significant economic losses around the world, the number of studies carried out to identify Bt proteins active against beetles is lower than those performed in Lepidopteran insect species. In the case of CBW, only a few studies have been performed to assess the insecticidal activity of either Bt strains or individual Bt insecticidal proteins. Therefore, more research is needed to determine which Bt proteins are able to effectively control CBW.

The application of Bt toxins to control *A. grandis* in cotton crops, especially as a transgenic plant, is of considerable agronomic interest, since this biotechnological tool is promising for reducing production and yield losses and containing CBW infestation, which represents a major problem for the cotton farmers in the Americas. Herein, in the present work, four Bt proteins known to be toxic to Coleopterans have been tested for the first time on *A. grandis* larvae. Two of the proteins tested belong to the 3-domain toxin class (Cry1Ba and Cry7Ab), and the other two mixed proteins (Mpp23Aa and Xpp37Aa, previously known as Cry23Aa and Cry37Aa, respectively) to the Mpp/Xpp class. Specifically, these last two proteins are expressed in a single operon and are mentioned as Mpp23Aa/Xpp37Aa. Therefore, results from this study could assess the potential of these proteins to effectively control CBW in the field.

## 2. Results

The survival of newly hatched larvae of *A. grandis* was affected by the ingestion of an artificial diet containing the different Bt proteins (Figure 1). The decreased survival of Bt-treated individuals increased with the increase in Bt protein concentration. For example, the analysis of the number of dead larvae after treatment at the highest dose (100 μg·g^−1^) was significantly different from that in the untreated group (one-way ANOVA, n = 269, F = 78, *p* < 0.0001). At this dose, the Cry1Ac, Cry1Ba, and Cry7Ab proteins had similar effects (Tuckey’s post-test, *p* > 0.99 for any combination), but the results were statistically different from those regarding the effect of Mpp23Aa/Xpp37Aa (Tuckey’s post-test, *p* > 0.0001). Cry proteins showed corrected mortality below 50%, which shows the low susceptibility of these insects to these proteins. In contrast, when the toxicity of the Mpp23Aa/Xpp37Aa proteins was tested at the same concentrations, 100% mortality was observed.

Moreover, the death of all larvae tested was found in all assayed concentrations, even at the lower level (1.56 μg·g^−1^). Based on these results, Mpp23Aa/Xpp37Aa proteins were selected for additional bioassay testing with lower toxin concentrations (ranging from 0.13 to 4 μg·g^−1^) to obtain dose–response data suitable for analysis. An example of the effect of these proteins can be observed in Figure 2.

The probit analysis was applied to the data obtained, estimating the bioassay parameters (Table 1). In addition, the analysis provided some extrapolated data of lethal concentration (LC) parameters useful for toxicity comparisons. As expected, Cry1Ac protein (described as a Coleopteran non-toxic) was marginally toxic (indicated by a high LC_50_ value). Based on the LC_50_ values obtained, Cry1Ba and Cry7Aa were slightly more toxic than Cry1Ac; however, the test of the hypothesis that slopes and intercepts are the same, discarded significant differences (χ^2^ = 7.04, degrees of freedom, d.f. = 4).

On the other hand, the Mpp23Aa/Xpp37Aa proteins showed significantly different toxicity dynamics (slope parameter) to *A. grandis* than to the Cry proteins tested, according to the previous test applied (χ^2^ = 532, d.f. = 2). The Mpp23Aa/Xpp37Aa toxicity was higher than that of the Cry proteins tested, since they provided a higher response to the concentration (a slope value higher), and the LC_10_ and LC_50_ values were lower. Only the LC parameters ratio (LC Cry1Ac protein/LC Mpp/Xpp proteins) could be provided, since the slopes were not parallel between Mpp/Xpp and the Cry proteins. The ratio of LC_10_ and LC_50_ values were 25 and 1761, respectively.

The concentration of Mpp/Xpp proteins required to kill 90% of the treated larvae (LC_90_), which is significant for pest control, was estimated as 0.4 μg·g^−1^ (fiducial limits, FL_95_ = 0.32–0.56), increasing the evidence of its high toxicity.

## 3. Discussion

Although *A. grandis* is currently one of the most devastating cotton insect pests in Brazil and other cotton crop-producing countries, there is little information about which Bt insecticidal proteins are effective against this insect pest [12,13,14]. The toxins usually described as active against insects of the order Coleoptera are those of the Cry3, Cry7, Cry8, Cry10, and Gpp34/Tpp35 classes [9,14], although other classes can be toxic as well, including the Cry1 class. Particularly, the Cry1Ba6 [16], Cry8Ka5 [17], Cry1Ia [18,19], Cry1Ia12 [20,21], and Cry10Aa [22] proteins have already been proven as toxic to CBW larvae in bioassays using purified recombinant protein or virus-infected insect extracts, while the Cry1Ia12 [20], Cry1Ia [19], and Cry10Aa [11,13] proteins were also confirmed to show some toxic activity when used in transgenic cotton lines.

It is important to highlight that studies evaluating the toxicity of single insecticidal proteins from Bt are required to show which proteins are effective against *A. grandis*. Furthermore, additional studies indicating whether it is suitable to combine the selected proteins in the same transgenic crop will be desirable to avoid the occurrence of cross-resistance among these proteins [23].

Results from the selective bioassay showed that although the survival of the larvae was affected after the ingestion of an artificial diet containing Cry1Ac, Cry1Ba, and Cry7Ab proteins, the highest dose used (100 µg·g^−1^) caused less than 50% mortality. The results obtained indicate that these proteins have low toxicity towards *A. grandis* larvae. The Cry1Ac is a protein known to be toxic to lepidopteran agricultural pest species such as *Helicoverpa zea*, *Heliothis virescens* [24], *Helicoverpa armigera* [25], *Anticarsia gemmatalis*, and *Chrysodeixis includens* [26], among others. For Coleopterans, although toxicity has been reported for some species [27], studies demonstrating high susceptibility to Cry1Ac are rare. Therefore, given the expression levels of Cry1Ac in commercial varieties, with a maximum of 7 and 20 µg·g^−1^ of fresh and dry leaf tissue, respectively [28,29], our results indicate that these current commercial varieties would not be able to control *A. grandis*.

Cry1Ba was included in this study because it has been described as a protein that has dual activity (toxicity to both Lepidopteran and Coleopteran insects) [14,30]. A previous study that tested the toxicity of Cry1Ba on *A. grandis* established an LC_50_ of 380.8 µg·g^−1^ [16], which is in line with our results. Interestingly, the authors reported that the toxicity of the original Bt strains that contained the *cry1B* gene was higher, suggesting that other virulence factors were causing the toxicity of these strains on *A. grandis*.

The Cry7 group is known to be poisonous to different Coleopteran insect species [10,14]. Here, we showed that the mortality of *A. grandis* caused by Cry7Ab was not high (40% as a mean in response to 100 µg·g^−1^), suggesting that this protein is not active against this insect pest. Similarly, *Anomala corpulenta* and *Pyrrhalta aenescens* larvae were not affected after ingestion of protein Cry7Ab3, and low susceptibility to Cry7Ab was observed in *Acanthoscelides obtectus* [30,31]. However, Cry7Ab was highly toxic to *Xylotrechus arvicola* and *Henosepilachna vigintioctomaculata* larvae, respectively [31,32].

On the other hand, Mpp23Aa/Xpp37Aa were extremely active against *A. grandis* larvae, with an estimated LC_50_ = 0.18 µg·g^−1^ (Table 1). These two proteins were described as toxic to *A. grandis* in a previous work [33], although this is the first study that establishes LC_50_ values. The toxic effect of Mpp23Aa/Xpp37Aa was also observed on larvae of *Tribolium castaneum* [34], *Cylas* spp. [35], and *Xylotrechus arvicola* [32], suggesting that these proteins are able to control different Coleopteran pests.

The Mpp23Aa/Xpp37Aa proteins were first described as a binary toxin based on the fact that both genes were located in the same operon. However, recently, it was shown that these proteins were toxic for *C. puncticollis* larvae when tested individually [36]. In our study, we have tested the toxicity against *A. grandis* larvae by using a *B. thuringiensis* strain that expresses both proteins; thus, we cannot determine whether both proteins are required to exert their toxic effect. Further research will be required to test whether both proteins are required for the toxic effect against *A. grandis* larvae.

Concentrations of some Cry proteins considered viable for introgression in cotton range up to about 20 µg·g^−1^, since the levels of protein expression in plants differ between flower buds and leaf tissues, ranging from approximately 3.0 to 19.0 µg·g^−1^ of fresh tissue [10,13]. Thus, according to our results, the potentially possible control of *A. grandis* by Mpp23Aa/Xpp37Aa toxins can be satisfactory. More experiments will be conducted to deepen the understanding of the toxicity and mode of action of this protein complex on *A. grandis*; however, the results presented here may be useful to outline several lines of research.

## 4. Materials and Methods

### 4.1. Bt Proteins Preparation

The source and preparation of the Bt proteins (Cry1Ba, Cry7Ab, and Mpp23Aa/Xpp37Aa) tested in this study were the *B. thuringiensis* strains described by Rodríguez-González et al. [30]. An exception was Cry1Ac, but it was prepared in the same way. In detail, Cry1Ac and Cry7Ab proteins were obtained from the *B. thuringiensis* HD-73 and HD867 strains, respectively, provided by the Bacillus Genetic Stock Center, USA. Cry1Ba was obtained from the recombinant *B. thuringiensis* strain EG11916 (Ecogen, Inc., Langhorne, PA, USA) and Mpp23Aa/Xpp37Aa were obtained from the EG10327 strain (Ref. No. NRRL B-21365) obtained from the Agricultural Research Culture Collection, Northern Regional Research Laboratory (NRRL), USA, respectively.

The *Bacillus thuringiensis* strains were grown in CCY medium supplemented with the appropriate antibiotic for 48 h at 29 °C, under constant agitation. Spores and crystals were separated by centrifugation at 16,000× *g* for 10 min at 4 °C and then washed three times with 1 M NaCl, 10 mM ethylenediaminetetraacetic acid (EDTA), and twice with 10 mM KCl. The final pellets were frozen using liquid nitrogen and lyophilized. The presence of the proteins in the lyophilized powder was determined using sodium dodecyl sulfate 12% polyacrylamide gel electrophoresis (SDS-PAGE) (Appendix A).

### 4.2. Rearing of Anthonomus grandis

Approximately 500 adult boll weevils collected from cotton fields in the region of Rondonópolis City, Mato Grosso, Brazil, were placed in cages at a 1:1 male–female ratio and maintained at 27 °C, under 40% relative humidity, and with a photoperiod of 12:12 h light:dark. In small, cube-shaped cages, the insects were fed an artificial diet [37].

The insect cages were serviced every two days, when the artificial diet was replaced by a fresh one, and insect eggs were collected. The eggs were superficially disinfected with a 20% copper sulfate and 0.2% benzalkonium chloride solution and separated from impurities such as diet leftovers and feces. Then, they were incubated in Petri dishes lined with filter paper and moistened with sterile water until hatching, and the newly hatched larvae (<48 h) were used for the bioassays.

### 4.3. Bioassays of A. grandis with Bt Proteins

The bioassays were based on the method of incorporating the test substances into an artificial diet [16]. The proteins, supplied as a lyophilized powder, were resuspended in sterile deionized water, and the concentration in this solution was determined by the Bradford method, using bovine serum albumin (BSA) as a standard. The aliquots required to obtain the doses corresponding to concentrations of 100; 50; 25; 12.5; 6.25, 3.13, and 1.56 µg·g^−1^ were incorporated into the artificial CBW diet in a total volume of 30 mL. After blending, the mixture was poured into Petri dishes (Ø 90 × 15 mm) for solidification, and then 35 newly hatched *A. grandis* larvae were placed at equal distances on the diet surface. The control for natural mortality consisted of an artificial diet, without the protein addition. The plates were incubated at 27 °C, with 40% relative humidity, and a photoperiod of 12:12 h light:dark for seven days, at which time the number of live and dead larvae on each plate was counted. All assays were performed in triplicate.

Mpp23Aa/Xpp37Aa proteins that induced very high mortality (100%) at the lowest dose (1.56 µg·g^−1^) were selected for an additional bioassay at concentrations of 4, 2, 1, 0.5, 0.25, and 0.13 µg·g^−1^, according to the methodology described above.

### 4.4. Data Analysis

Mortality results for the figure were corrected based on mortality observed in the control using Abbott’s formula [15], obtaining the mean value and the standard error of the mean. All data analyses were performed with uncorrected mortality data. Differences in dead larvae among different protein treatments at 100 µg·g^−1^ were tested by one-way ANOVA, followed by a Tukey’s post-test, considering *p* < 0.05 as a significant difference. GraphPad Prism 7.0 (GraphPad Software, La Jolla, CA, USA) was used.

Bioassay data were analyzed through probit analysis [38] using the POLO-PC software program (LeOra Software, Berkeley, CA, USA, 1987) in order to obtain the toxicological parameters and to compare the obtained results.

## Figures and Tables

**Figure 1 toxins-15-00055-f001:**
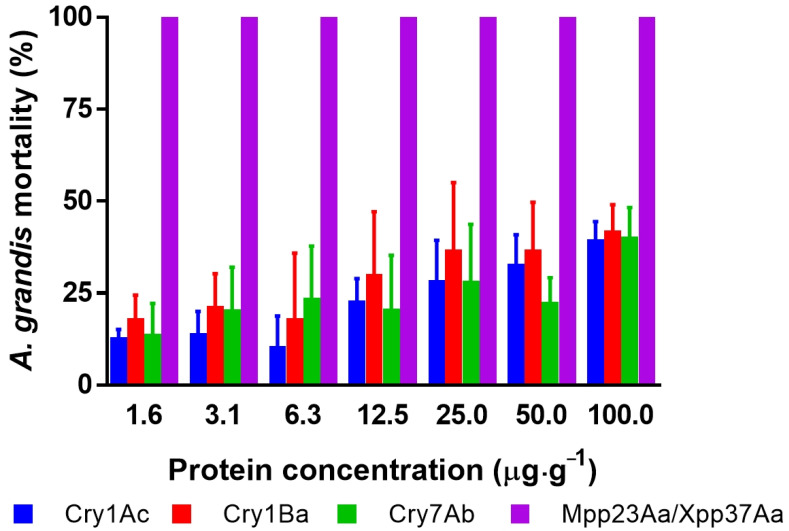
Corrected mortality [15] of *A. grandis* larvae after seven days of protein intake in an artificial diet at seven concentrations of different *B. thuringiensis* Cry proteins. Error bars represent the mean standard error. The error bars for Mpp23Aa/Xpp37Aa are not shown because of the death of all treated larvae in all replicates.

**Figure 2 toxins-15-00055-f002:**
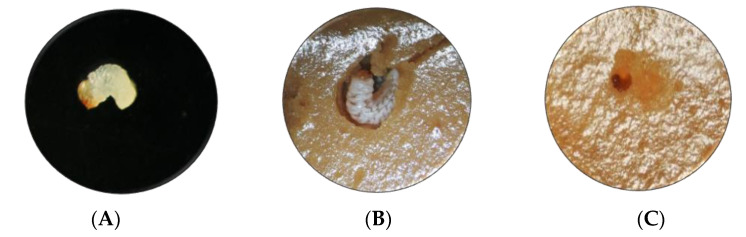
Effect of Mpp23Aa/Xpp37Aa on *A. grandis* larvae with 0.5 μg·g^−1^ treatment. (**A**) Neonate larva at the start of the treatment (1 mm size), (**B**) larva on control untreated media after 7 days of treatment (5 mm size), (**C**) larva on Mpp/Xpp treated media after 7 days of treatment (1 mm size).

**Table 1 toxins-15-00055-t001:** Parameters estimated by probit analysis from the bioassays with *A. grandis* larvae tested with several Bt proteins. Toxin concentrations are given in μg of protein per·g^−1^ of artificial diet.

Bt Protein	n	Slope (±SE)	LC_10_ (FL_95_)	LC_50_ (FL_95_)	χ^2^ (d.f.)
Cry1Ac	841	0.58 (0.16)	2.01 (0.04–7.70)	317 (188–3378) *	2.08 (5)
Cry1Ba	840	0.43 (0.11)	0.33 (0.01–1.84)	293 (92–4486) *	2.05 (5)
Cry7Aa	841	0.35 (0.12)	0.38 (0.01–3.05)	1598 (233–ND) *	4.60 (5)
Mpp23Aa/Xpp37Aa	735	3.80 (0.47)	0.08 (0.04–0.12) *	0.18 (0.13–0.22)	4.43 (4)

n = number of evaluated larvae; slope = dose–response quickness; SE = standard error of the mean; LC_10_ and LC_50_ = concentration that killed 10% and 50%, respectively; FL_95_ = 95% fiducial limits; d.f. = degrees of freedom; ND = not determined; * value interpolated because they were out of the range of assayed concentrations.

## Data Availability

The datasets generated and/or analyzed during the current study are available from the corresponding author upon reasonable request.

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
