# Peer review of "Mpp23Aa/Xpp37Aa Insecticidal Proteins from *Bacillus thuringiensis* (Bacillales: Bacillaceae) Are Highly Toxic to *Anthonomus grandis* (Coleoptera: Curculionidae) Larvae"

_toxins, 2023, doi:10.3390/toxins15010055_

Round 1

Reviewer 1 Report

This manuscript provides us a quantitative toxicity data of the Mpp23Aa/Mpp37Aa insecticidal proteins from Bt, which is important to support subsequent studies, especially technique development to control the cotton boll weevil. However, there are some comments on this communication.

1.      In Table 1., it is a difference between the parameters in the first line and those in the table caption. LC10 and LC50 Vs LC50 and LC90

2.      Although we believe the material is correct, it is desirable to accurately identify the insecticidal proteins besides checking with SDS-PAGE.

3.      Some spelling or typographical errors should be corrected. For example, in line 247, ”1,56” should be “1.56”.

Author Response

We thank the reviewer for his/her comments that improve the paper.

  1. In Table 1., it is a difference between the parameters in the first line and those in the table caption. LC10 and LC50 Vs LC50 and LC90

It has been corrected

  1. Although we believe the material is correct, it is desirable to accurately identify the insecticidal proteins besides checking with SDS-PAGE.

We added a figure with a gel with the proteins as supplementary material (line 229 of the revised manuscript)

  1. Some spelling or typographical errors should be corrected. For example, in line 247, ”1,56” should be “1.56”.

The text was thoroughly revised and some spelling and typographical errors have been corrected along the text.

Reviewer 2 Report

General remarks

The paper entitled “Mpp23Aa/Mpp37Aa insecticidal proteins from Bacillus thuringiensis (Bacillales: Bacillaceae) are highly toxic to Anthonomus grandis (Coleoptera: Curculionidae) larvae” provides new data for using insecticidal proteins in order to control the cotton pest. The impact of different proteins with various concentrations included in the diet of coleopteran larvae was evaluated. The obtained data sustain the idea that Mpp23Aa/Mpp37Aa protein is a promising alternative for controlling A. grandis beetle pest.

Comments

The approached subject is interesting especially in the actual context of the continuous efforts in developing new insecticides with significant effect while used in reduced amounts.

The aims of the study are clearly expressed.

The obtained results are concise, clearly presented and discussed.

The experimental program is described in such manner that it can be easily applied.

Therefore, with some minor corrections (grammar errors to correct), the paper can be published. 

Author Response

We thank the reviewer for his/her comments that improve the paper. We revised the manuscript to correct the typographical and grammatical errors.

Reviewer 3 Report

Overall the manuscript is interesting and can contribute significant knowledge in eliminating pest in cotton industry.

Title: accepted

Introduction

Well elaborate with the latest references

But please provide the current treatments use to against the pest

Results. Clear with scientific merit

Discussion

good

Materials and methods

Accepted

Referecences

Please find and make sure all or at least 75% of references should be within 5 years

Author Response

We thank the reviewer for his/her comments that improve the paper.

  1. please provide the current treatments use to against the pest

We have included text with the requested information in the introduction section (lines 51-59 of the revised manuscript).

  1. Please find and make sure all or at least 75% of references should be within 5 years

We have revised the references to try to update them, however the field is built with long-term studies and sometimes in singular works. In this way, we prefer to give credit to the citations in the references to the original works instead of the reviews that compile them. Only few references have been modified.